# GDSL Esterase/Lipase GELP1 Involved in the Defense of Apple Leaves against *Colletotrichum gloeosporioides* Infection

**DOI:** 10.3390/ijms241210343

**Published:** 2023-06-19

**Authors:** Zhirui Ji, Meiyu Wang, Shuwu Zhang, Yinan Du, Jialin Cong, Haifeng Yan, Haimeng Guo, Bingliang Xu, Zongshan Zhou

**Affiliations:** 1College of Plant Protection, Gansu Agricultural University, Lanzhou 730070, China; xinyu_jzr@163.com (Z.J.); zhangsw704@126.com (S.Z.); 2Research Institute of Pomology, Chinese Academy of Agricultural Sciences, Xingcheng 125100, China; wang903108175@163.com (M.W.); duyinan0312@163.com (Y.D.); sprinctt@foxmail.com (J.C.); yanhaifeng50@163.com (H.Y.); ghm05284@163.com (H.G.)

**Keywords:** GDSL esterases/lipases, Fuji, resistance, Glomerella leaf spot

## Abstract

GDSL esterases/lipases are a subclass of lipolytic enzymes that play critical roles in plant growth and development, stress response, and pathogen defense. However, the GDSL esterase/lipase genes involved in the pathogen response of apple remain to be identified and characterized. Thus, in this study, we aimed to analyze the phenotypic difference between the resistant variety, Fuji, and susceptible variety, Gala, during infection with *C. gloeosporioides*, screen for anti-disease-associated proteins in Fuji leaves, and elucidate the underlying mechanisms. The results showed that GDSL esterase/lipase protein GELP1 contributed to *C. gloeosporioides* infection defense in apple. During *C. gloeosporioides* infection, *GELP1* expression was significantly upregulated in Fuji. Fuji leaves exhibited a highly resistant phenotype compared with Gala leaves. The formation of infection hyphae of *C. gloeosporioides* was inhibited in Fuji. Moreover, recombinant His:GELP1 protein suppressed hyphal formation during infection in vitro. Transient expression in *Nicotiana benthamiana* showed that GELP1-eGFP localized to the endoplasmic reticulum and chloroplasts. GELP1 overexpression in GL-3 plants increased resistance to *C. gloeosporioides*. *MdWRKY15* expression was upregulated in the transgenic lines. Notably, GELP1 transcript levels were elevated in GL-3 after salicylic acid treatment. These results suggest that GELP1 increases apple resistance to *C. gloeosporioides* by indirectly regulating salicylic acid biosynthesis.

## 1. Introduction

GDSL-type esterase/lipase proteins (GELPs) belong to the SGNH hydrolase superfamily and contain a conserved GDSL motif at the N-terminus. Over 1100 members of the GDSL family of esterases/lipases have been found in 12 fully sequenced plant genomes [1], and they have become highly attractive topics because of their recently discovered properties and functions. Several plant GELPs have been isolated, cloned, and characterized. Members of the GELP family are primarily involved in the regulation of plant development, morphogenesis, secondary metabolite synthesis, and defense response [2,3]. In *Arabidopsis*, *GELP77* functions in pollen dissociation and fecundity [4], secondary metabolism [5,6], and immunity [7]. The overexpression of *AtGDSL1* promotes resistance to *Sclerotinia sclerotiorum* in oilseed rape by modulating salicylic acid (SA)–dependent and jasmonic acid (JA)–dependent pathways, thereby increasing phosphatidic acid accumulation and activating downstream stress response pathways following *Sclerotinia* infection [8]. OsGLIP1 and OsGLIP2 negatively regulate rice defense by modulating lipid metabolism, providing novel insights into lipid function in plant immunity [9]. In addition, GDSL lipases play roles in plant development, seed germination, morphogenesis, and pathogen defense responses [10,11]. However, the enzymatic or biological functions of most GELPs remain unclear.

Glomerella leaf spot of apple (GLSA), a destructive fungal disease caused by *Colletotrichum gloeosporioides*, severely affects apple quality and yield [12,13,14]. Since GLSA was first reported and identified to be caused by *Glomerella cingulate* (anamorph: *C. gloeosporioides*) in the USA [15,16], it has affected the apple industry in many countries worldwide [15,17,18]. In China, GLSA was first discovered in the Gala apple orchard in Jiaozuo, Henan Province. The disease occurs at a rapid rate during the growing season, causing severe defoliation in 20% of the leaves within 2–3 days [17,19]. In recent years, this disease has become serious in major apple–producing areas and has caused serious economic losses to the apple industry [20,21,22]. Currently, GLSA is controlled through the spray application of the fungicides dithiocarbamate, tebuconazole, and prochloraz; however, this approach decreases food safety, pollutes the environment, and increases production cost [23,24]. The Fuji variety group exhibits a high disease resistance phenotype [25]. Clarifying the molecular mechanisms underlying disease resistance is important for controlling the disease through molecular breeding. In recent years, some studies have focused on the molecular mechanisms underlying the interaction between pathogens and host plants, and a small number of effectors have been identified in *Colletotrichum* [24,26].

When pathogen–associated molecular patterns or effectors are recognized by plants during an infection, cellular signaling inside the cell may be activated, leading to the influx of calcium, the production of reactive oxygen species (ROS), the activation of mitogen-activated protein kinases (MAPKs), and the induction of defense genes [27]. Studies have confirmed that WRKY transcription factors exert their disease-resisting properties via the MAPK signaling pathway. *Arabidopsis* WRKY22/29 is induced by a signaling cascade which includes AtMEKK1, AtMKK4/5 and AtMPK3/6, thereby enhancing fungal resistance in the plants [28]; Similarly, AtWRKY25/33 is induced by a cascade which includes AtMEKK1, AtMEK1/AtMKK2 and AtMPK4 to regulate the expression of various downstream defense genes which partici-pate in SA inhibition and JA activation [29]. Resistance–related genes have also been identified in apples (*Malus domestica Borkh*) [30,31,32]. Zhang et al., reported that *miRln20*, a GLSA–responsive gene, negatively regulates GLSA resistance [30]. WRKY transcription factors play key roles in plant resistance to various pathogens as important targets of MAPK signaling cascade members. *MKK4-MPK3-WRKY17*–mediated SA degradation increases Glomerella leaf spot susceptibility in apples [31]. *MdWRKY100* positively regulates plants’ resistance against *C. gloeosporioides* [32]. Additionally, MPK3/MPK6 signaling and downstream WRKY transcription modulate GDSL lipase gene expression [33]. Plant GDSL esterases/lipases (GLIP1 and GLIP3) from *Arabidopsis* contribute to plant resistance against *Botrytis cinerea* [33]. GLIP1 regulates systemic immunity by modulating ethylene (ET) signaling components [34]. GLIP2 is involved in the resistance against *Erwinia carotovora* by negatively regulating auxin signaling [35]. Consistent with previous findings, the regulatory model of AtWRKY33 and MdWRKY17 that participate in fungal disease resistance downstream of the MAPK cascade was constructed (Appendix A) [31,36,37,38,39,40,41,42]. However, the signaling pathways through which GDSL esterases/lipases confer apple resistance against *C. gloeosporioides* and their application in breeding *C. gloeosporioides*–resistant apple cultivars remain to be identified.

To explore the resistance mechanisms of Fuji, we aimed to analyze the phenotypic difference between the resistant variety, Fuji, and the susceptible variety, Gala, during infection with *C. gloeosporioides* and screen for disease-resistance–related proteins in Fuji leaves. Specifically, we characterized the GELP1 protein in apple leaves and determined its role in regulating apple resistance to the *C. gloeosporioides* wild-type strain W16.

## 2. Results

### 2.1. Fuji Leaves Resist C. gloeosporioides Infection

Infection characteristics of the resistant variety, Fuji, and susceptible variety, Gala, were detected. After 4 days of inoculation with *C. gloeosporioides*, large disease spots appeared on Gala leaves, but none appeared on Fuji leaves (Figure 1A). The disease index of Gala was up to 76.1%, whereas that of Fuji was 0% at 4 days postinoculation (dpi; Figure 1A). The infection structures of W16 were observed 24 h postinoculation (hpi) and reached 83.3% in Gala leaves but only 51.2% in Fuji leaves (Figure 1B). At 72 hpi, the formation of infection hyphae in Fuji leaves was 0%, whereas that in Gala was 53.1% (Figure 1B). These results indicated that Fuji produced certain structures or substances that impede the formation of infection hyphae and prevent W16 from completing leaf infection and disease.

Host plants recognize pathogens and produce high levels of defense genes, ROS generation, and corpus callosum deposition to inhibit pathogen infections [43,44]. We detected H_2_O_2_ accumulation at the infection site using 3′-diaminobenzidine (DAB) staining. Histological examination revealed that H_2_O_2_ accumulation was dramatically higher at the infection site in Fuji leaves than in Gala leaves (Figure 1C). *Callose synthase 2*-like was also upregulated in Fuji leaves during infection (Figure 1D). The amounts of callose deposits were higher in Fuji leaves infected with the deletion mutant of effector Sntf2 than in those infected with W16 (Figure 1E). Sntf2 has previously been shown to play an important role in inhibiting plant defense responses to *C. gloeosporioides infection* [45]. These results indicated that *C. gloeosporioides* induces the plant immune defense response during the infestation of Fuji apple leaves.

### 2.2. GELP1 Expression Is Upregulated in Fuji during Infection

GDSL lipases/esterases play important roles in plant development and disease defense [9,46]. The screening of the Fuji Y2H library revealed a potential interacting protein of the virulence effector Sntf2 that encodes a GDSL lipase/esterase-like protein (Gene ID: 103451100), which we named GELP1 (Figure 2A). The transient co-expression showed that the signal of GELP1–eGFP was co-localized in the chloroplasts with the signal of Sntf2^Δsp^-TagRFP (Appendix A). However, the qRT–PCR results showed that *GELP1* expression was significantly upregulated in Fuji at 24 hpi (Figure 2B). To determine the role of *GELP1* in the infection, we analyzed its expression in Gala at different stages of infection. The results showed that *GELP1* was downregulated during the infection (Figure 2C). Semi-quantitative RT–PCR analysis showed that the transcription levels of *GELP1* were higher in Fuji leaves than in Gala leaves (Figure 2D). *GELP1* was induced by *C. gloeosporioides *infection in Fuji (Figure 2D). These results indicated that *GELP1* participated in the immune defense response of apple.

### 2.3. GELP1 Is a GDSL Lipase/Esterase-Like Protein

NCBI BlastP analysis revealed that another GDSL lipase/esterase–like gene (*GELP2*) in the Golden Delicious genome (Gene ID: 103434228) is 93.37% similar to *GELP1* (Appendix A). Golden Delicious was also susceptible to GLSA (Appendix A). *GELP1* and *GELP2* are encoded as 362-aa proteins containing a GDSL domain, and hydrolase activity acts on ester bonds, as predicted by InterPro 85.0 (Figure 3A). RT–PCR analysis showed that the expression of *GELP2* in Gala and Fuji leaves was similar to that of *GELP1* (Figure 3B). However, *GELP2* expression was upregulated in Gala leaves at 12 hpi and downregulated at 24 hpi. Furthermore, cis–element analysis revealed that the promoter region of *GELP1* contained defensive and stress–responsive cis–elements and SA–responsive cis–elements (Appendix A). *GELP2* is mainly involved in light-responsive cis-elements but also contains abscisic acid- and methyl jasmonate–responsive cis–elements (Appendix A). Thus, GELP2 and GELP1 may respond to different stresses.

SignalP analysis revealed that GELP1 contains an N–terminal signal peptide (SP) (1–24 aa). To determine GELP1 localization, we constructed *GELP1* (with or without SP) and GFP–fused expression vector and expressed the GELP1–eGFP protein using Agrobacterium-mediated transient expression in *Nicotiana benthamiana* leaves. The GELP1–eGFP fluorescent signal accumulated in the endoplasmic reticulum (ER) and chloroplasts (Figure 3C). However, the GELP1 signaling peptide was not associated with its localization in *N. benthamiana* (Figure 3C). Therefore, exploring the localization of GELP1 in the leaves of Fuji apple and the function of the protein is essential.

### 2.4. GELP1 Suppresses the Formation of Infection Structures

To determine the antimicrobial activity of GELP1 against *C. gloeosporioides*, we expressed recombinant His–GELP1 fusion proteins in *Escherichia coli* M15 cells (Figure 4A). The purified His–GELP1 was determined through Western blotting (Figure 4A). No significant difference in appressorium formation was observed between the groups treated with His–GELP1 and His (the blank control) at 12 h of co–incubation (Figure 4B). However, the formation of infection hyphae was suppressed by His–GELP1 at 24 h of co–incubation (Figure 4B). We also inoculated the incubated conidia on GL–3 leaves to analyze whether His–GELP1 induces local and systemic plant resistance. The co–incubation of apple leaves for 24 h inhibited the germination and formation of *C. gloeosporioides* and reduced the formation of infection hyphae compared with the control (Figure 4C). These results suggested that GELP1 inhibited the formation of infection hyphae.

### 2.5. GELP1 Enhances Plant Resistance to C. gloeosporioides

To further analyze the function of *GELP1*, we constructed an overexpression transgenic plant. All transgenic lines were screened using kanamycin. Transgenic lines overexpressing *GELP1* (OE–GELP1) were confirmed using qRT–PCR and RT–PCR. *GELP1* expression was significantly higher in the transgenic lines than in GL–3 (Figure 5A) and Fuji (Figure 5B). Compared with GL–3, the transgenic lines showed a similar phenotype on Murashige and Skoog medium (Figure 5C) and showed good growth rate and redifferentiation. The pathogenicity of W16 in OE–GELP1 leaves was also investigated. The disease index of the overexpression strain was significantly reduced compared with that of GL–3 (Figure 5D). In addition, after inoculation with spore suspension for 72 h, disease spots appeared on GL–3, whereas the transgenic plants had no disease spots, indicating that the disease resistance of OE–GELP1 was significantly improved (Figure 5E).

No significant difference in appressorium formation was found between W16 and GL–3 leaves at 24 hpi (Figure 6A). However, the infectious phase of OE–GELP1 was significantly suppressed compared with that of GL–3 (Figure 6A). In addition, aniline blue staining showed higher amounts of callose deposits in the infection sites of OE–GELP1–1 and OE–GELP1–2 leaves than in those of GL–3 leaves at 24 hpi (Figure 6B). qRT–PCR analysis showed that the expression of *Callose synthase 2-like* was higher in OE–GELP1–1 leaves than in GL–3 leaves (Figure 6B). In addition, MdLAPX6 and MAPKKK3 were upregulated in OE–GELP1–1 and Fuji leaves (Figure 6C). MAPKKK3 encodes a component of the MAPK pathway, and MAPKKK3/MAPKKK5 is required for basal resistance to Pto-DC3000 [47]. These results suggested that GELP1 overexpression increased resistance to *C. gloeosporioides* in GL–3.

### 2.6. GELP1 Was Related with SA Accumulation

Exogenous SA application in apple leaves enhances resistance to GLSA [48]. In the present study, we sprayed SA onto Fuji and Gala leaves. Under SA stress, *GELP1* expression was upregulated in Gala leaves (Figure 7A) but showed no change at 0 and 24 h in Fuji leaves (Figure 7A). However, the *GELP1* expression in Gala leaves was similar to that in Fuji leaves at 24 h under SA stress (Figure 7A). In the present study, *MdWRKY15* expression was only slightly upregulated in OE–GELP1–1 leaves compared with GL-3 leaves at 24 hpi, and it was also upregulated in Fuji leaves at 24 hpi (Figure 7B). The transcription factor *MdWRKY15* activates the SA synthetase *MdICS* [49]. Furthermore, *WRKY17* expression was downregulated in OE–GELP1–1 leaves than in GL-3 leaves at 24 hpi and upregulated in GL–3 leaves at 24 hpi (Figure 7C). Moreover, *WRKY100* expression was only slightly upregulated in OE–GELP1–1 leaves (Figure 7D). *MdWRKY100* and *MdWRKY17* have been linked to SA accumulation [31,32]. Reducing SA accumulation helps apples infected with *Colletotrichum fructicola* [31]. These results indicated that *GELP1* promoted plant resistance to *C. gloeosporioides* by increasing SA accumulation.

## 3. Discussion

*C. gloeosporioides* causes semi-in vitro nutrient infection on the susceptible cultivar, Gala, and mismatch or latent infection in vitro on the resistant cultivar, Fuji [50]. In the present study, in infected Fuji leaves, the formation of infection hyphae was inhibited, and the H_2_O_2_ accumulation and callose deposition increased. In addition, we identified a GDSL esterase/lipase At5g33370-like protein, GELP1, and found that its expression was significantly upregulated during Fuji infection. GDSL plays a role in plant growth and development, disease resistance, and lipid synthesis. *Arabidopsis* At5g33370 encodes the protein AtGELP95 (CUS2), which is involved in cutin polymer formation and cuticle permeability [51]. Tomato lipase GDSL1 is embedded in the cuticle matrix outside the epithelial cells and is associated with epidermal thickness, cuticle density, and resistance [52,53]. The GDSL esterase/lipase OsGELP78 modulates lipid metabolism to regulate rice resistance to disease [54]. In the present study, we found that GELP1 was highly similar to the LTL1–like GDSL esterase/lipase protein, whereas *Arabidopsis thaliana* Li-tolerant lipase 1 (AtLTL1) expression was activated by SA, suggesting that this lipase may also be involved in pathogen defense [55]. GELP1 overexpression increased apple resistance to *C. gloeosporioides* in GL–3. Callose deposits increased, and the expression of *Callose synthase 2-like* and *MdLAPX6* was also upregulated. *Apxs* plays a role in scavenging ROS and protecting cells. In the present study, GELP1 is also involved in biotic stress responses.

The enzyme responsible for TAP in *E. coli* is a multifunctional GDSL esterase/lipase that functions as a thioesterase, protease, arylesterase, lysophospholipase, and esterase [56,57]. GDSL hydrolases have a flexible active site and switch conformation when different substrates are present, and some GDSL enzymes have a wide range of enzymatic activities, including esterase and protease activities in the same enzyme [58,59]. AtLTL1 and SlGDSL1 are orthologous extracellular acyltransferases in *Arabidopsis* and tomato, respectively, which catalyze the formation of line cuttings in oligomers using 2-MHG in vitro [52,60]. The recombinant GLIP1 protein possesses lipase and antimicrobial activities, which directly disrupt the integrity of fungal spores [9,61]. GLIP2 also possesses lipase and antimicrobial activities that inhibit the germination of fungal spores [62]. In the present study, recombinant GELP1 inhibited the formation of *C. gloeosporioides* infected hyphae. Plant GDSL lipase/esterase GLL25 is required for normal ER morphology and regular organelle distribution, and lipase/esterase activity is lost without a conserved active site of the GDSL motif [63].

Chloroplasts are important energy-converting cells in plants. In addition to photosynthesis, it plays a role in the synthesis of amino acids, fatty acids, plant growth substances, nucleotides, vitamins, and secondary metabolites [64]. Chloroplasts can be targeted as effectors, such as bacterial effectors HopI1, HopK1, and AvrRps4, which inhibit chloroplast-derived ROS or SA [65,66]. In the present study, a small fraction of GELP1-eGFP fluorescence was observed in the chloroplasts. Lipids are widely distributed in plants and are thought to be involved in regulating plant immunity [9]. DGDG are abundant galactolipids present in the thylakoid membranes and photosynthesis systems I and II [67]. Interestingly, DGDG contributes to the biosyntheses of NO and SA, which are involved in defense responses [6]. In the present study, WRKY genes were upregulated in the GELP1 overexpression transgenic lines, which was related to SA accumulation. We speculate that the chloroplast location is linked to SA biosynthesis.

The defense responses of plants to pathogens are regulated by networks of SA, JA, and ET signaling pathways [68]. In *Arabidopsis*, PRLIP1 induction depends on a functioning SA and ET signal transduction pathway [69]. GLIP1 responds to SA and functions in resistance to *Alternaria brassicicola* [61]. It directly targets MPK3/MPK6 and its downstream transcription factor WRKY33. *Arabidopsis* GDSL lipase 1 (GLIP1) requires ET signaling components for its expression and regulates systemic immunity via the positive (i.e., ERF1 activation) and negative (i.e., EIN3 degradation) feedback regulation of ET signaling [70]. In the present study, GELP1 overexpression indirectly upregulated the expression of WRKY genes, which are associated with SA generation. GELP1 expression was also upregulated by the external application of SA. In another study, the overexpression of the GDSL motif lipase/hydrolase-like protein GLIP1 or treatment with exudates from 35S: GLIP1 plants induce systemic pathogen resistance, and this effect has been associated with JA/ET and SA [61]. In the present study, GELP1 overexpression increased resistance to *C. gloeosporioides*. In our next work, we will investigate whether GELP1 can elicit a broad spectrum of resistance to multiple pathogens.

WRKY transcription factors play important roles in resistance to various fungal diseases in apples [71,72]. *MdWRKY100* positively regulates resistance to *C. gloeosporioides* infection [32]. RNAi-mediated silencing of *MdWRKY17* in Gala plants promotes GLS tolerance, and MdWRKY17 negatively regulates SA levels, leading to GLS susceptibility [31]. Other studies have shown that SA is critical for GLS tolerance [48]. In plants, SA levels are primarily determined by the balance between SA biosynthesis (mainly controlled by ICS1) and degradation [73]. The transcription factor *MdWRKY15* activates the SA synthetase *MdICS1* [49]. SA confers enhanced resistance to GLSA [48]. In the present study, GELP1 overexpression increased resistance to *C. gloeosporioides*. The expression of *MdWRKY15* and *MdWRKY100* was upregulated, whereas that of *MdWRKY17* was downregulated in the *GELP1* overexpression transgenic lines. These findings highlight that SA plays a role in regulating resistance to *C. gloeosporioides* infection and that *GELP1* contributes to this resistance by enhancing SA accumulation.

## 4. Materials and Methods

### 4.1. Materials and Growth Conditions

W16 was isolated and purified by our group [74] GL–3 (Royal Gala) plants were supplied by Professor Zhihong Zhang [75]. Appendix A lists the primers used in the present study. W16 was cultured on PDA agar plates at 25 °C in the dark. Transgenic lines overexpressing *GELP1* (OE–GELP1) and GL-3 plants were grown on Murashige and Skoog medium in a climate-controlled culture room at 25 ± 1 °C with a light/dark photoperiod of 16/8 h [75,76]. Seedlings of *N. benthamiana* were grown in a greenhouse at 22–25 °C and 75% humidity. Healthy leaves of Fuji and Gala were obtained from seedlings in our research orchard (Institute of Pomology of Chinese Academy of Agricultural Sciences, CAAS, Xingcheng, Liaoning Province, China).

### 4.2. Phenotypic Analysis

Fresh conidial suspensions were harvested from the surface of PDA cultures (9 cm-diameter plates) with 10 mL of sterile distilled water. The conidial suspensions were centrifuged at 5000 rpm, resuspended in normal saline, and then washed three times. For plant inoculation, fresh conidial suspensions (1 × 10^6^ conidia/mL) were sprayed onto the leaves. The inoculated apple leaves were grown at 28 °C with a humidity of 75%. The formation of fungal appressoria and infection hyphae in the leaves was assessed via light microscopy after discoloration by boiling with 95% anhydrous ethanol and transparency using chloral hydrate. Pathogenicity assays were performed by examining disease lesions at 3 dpi. The proportion of diseased lesions (lesion area/leaf area) was graded in accordance with the following scale: 1: ≤20%; 3: >20%, ≤40%; 5: >40%, ≤60%; 7: >60%, ≤80%; 9: >80%. The disease index was expressed as ∑ (number of lesions × associated lesion grade)/(total number of lesions × largest lesion grade).

### 4.3. Histochemical Assays

DAB staining was performed as previously described to evaluate H_2_O_2_ production in plants [77]. Samples of inoculated leaves were collected at different stages of infection. DAB oxidation leads to the formation of a yellow polymer at the site of H_2_O_2_ accumulation. Callose deposits were detected via aniline blue staining. Samples were obtained from WT–inoculated leaves, which were decolorized by boiling in 95% ethanol for 5 min, and then dipped in chloral hydrate to make them transparent. The processed leaf segments were stained with 0.05% aniline blue in 0.067 M K_2_HPO_4_ (pH 9.6) [77]. For microscopy, the samples were preserved in 40% glycerol. The experiments were conducted at 0, 24, 36, and 48 hpi.

### 4.4. RNA Extraction and RNA Analyses

Total RNA was isolated from apple leaves using an RNA Prep Pure Plant Kit (TianGen Biotech, Beijing, China) in accordance with the manufacturer’s protocol. First-strand complementary DNA (cDNA) was synthesized using a Two-Step RT–PCR SuperMix kit (TransGen Biotech). RT–PCR analysis of 100 ng total RNA was performed using RT-PCR SuperMix (TransGen Biotech) for the detection of GELP1 and MdUBQ. 5 The assay was conducted in triplicate. qRT–PCR was performed using the Tip Green qPCR SuperMix (TransGen Biotech). All of the qRT–PCR products were analyzed using Bio-Rad CFX Manager 3.0 software. Gene expression was measured by normalizing the expression values to those obtained from conidia or non-inoculated apple leaves using the β–tubulin gene as an endogenous reference. Each sample and the entire experiment were performed in three separate triplicates.

### 4.5. Yeast Two-Hybrid Assay

The coding sequences of *SNTF2* were cloned into pGBKT7-BD as bait, and the Matchmaker GAL4 system (Shanghai OE Biotech, Shanghai, China) was used to screen a library of cDNA constructed from RNA isolated from infected Fuji leaves. The screening was performed in accordance with the manufacturer’s instructions (Shanghai OE Biotech). The prey vector pGBKT7–GELP1 was constructed using the *GELP1* cDNA from Fuji leaves to confirm the interaction between Sntf2 and GELP1. The corresponding prey and bait vectors were co-transformed into the Y2HGold strain. The transformed yeast strains were grown on a medium (SD/–Leu/–Trp and SD/–Leu/–Trp/–His/–Ade) at 30 °C for 3 days to select the positive transformants.

### 4.6. Transient Expression Analysis in N. benthamiana

For the subcellular localization assay, *GELP1* was cloned into pGR35s–eGFP and transformed into *Agrobacterium tumefaciens* LBA4404. The transformed strains were resuspended in infiltration buffer (10 mM MgCl_2_, 10 μM AS, 10 mM 2–(N–morpholino) ethanesulfonic acid, pH 5.6) at OD_600_ = 0.3 and injected into 4-week-old *N. benthamiana* leaves. The empty vector pGR35s-eGFP served as a positive control. After 2 days of infiltration, the injected leaves were observed with an epifluorescence microscope (Leica TCS SP8). For GFP fluorescence detection, the excitation and emission wavelengths were 488 and 495–545 nm, respectively.

### 4.7. Protein Extraction and Western Blot Analysis

*GELP1* was cloned into the pQE30 vector and transformed into *E. coli* M15 cells. Transformed cell suspension (OD600 = 0.80) was cultured, and protein expression was induced by 100 μM isopyl β–D–thiogalactopyranoside at 180 rpm for 3 h at 37 °C. Induced samples were centrifuged at 10,000 rpm for 5 min, suspended with 1 × PBS (sodium phosphate buffer) (140 mM NaCl, 2.7 mM KCl, 10 mM Na_2_HPO_4_, 1.8 mM KH_2_PO_4_, pH 7.3) with lysozyme and proteinase inhibitor, and then disrupted via sonication. After centrifugation at 12,000 rpm for 30 min at 4 °C, the supernatant was purified using Ni-NTA beads (CWBIO, CW0894S) in accordance with the manufacturer’s protocol. The induced proteins were detected through SDS-PAGE and Coomassie Blue staining. For immune detection, the purified proteins were transferred onto polyvinylidene fluoride membranes and detected using the corresponding mouse anti-His antibody (1:1000; cat. no. HT501; TRANS, China) with secondary horseradish peroxidase-labelled goat anti-mouse IgG (1:2000; cat. no. A0216, Beyotime, Shanghai, China). Images were acquired using a Bio-Rad ChemiDoc imaging system.

### 4.8. Chemical Treatment

SA (Fluka) was dissolved in water (1 mM SA, 0.01% Tween 20) and sprayed onto the leaves (1 mL/plant). *GELP1* transcript levels were measured through RT–PCR at 0, 12, 24, 48, and 72 h after treatment with 1 mM SA. Leaves of uniform size to the control were sprayed with distilled water. The antimicrobial activity of GELP1 was determined by adding recombinant proteins (0.5, 1, and 3 μg) to 10 μL of conidia suspension in (1 × 10^6^), and the onion epidermis and leaf surface were inoculated at 25 °C. The formation rates of appressoria and infection hyphae were detected using light microscopy.

### 4.9. Generation of Transgenic GL–3 Plants with GELP1 Over-Expression

The *GELP1* coding sequence was amplified and inserted into the pRPHA vector, which was then used to transform LBA4404 cells. The transformed strains were used to generate the overexpression transgenic line OE–GELP1. The *Agrobacterium*–mediated transformation of GL–3 was conducted as previously described [75].

## 5. Conclusions

We identified the GDSL esterase/lipase protein GELP1 and found that its expression was significantly upregulated in Fuji during *C. gloeosporioides* infection. GELP1 accumulated in the ER and chloroplasts and inhibited the formation of infection hyphae. GELP1 overexpression increased resistance to *C. gloeosporioides*. In addition, GELP1 was involved in SA biosynthesis. These results indicate that *GELP1* contributes to *C. gloeosporioides* resistance by enhancing SA accumulation. This study provides insights into the mechanisms underlying GLSA resistance in apple and may serve as a basis for developing novel strategies to control the disease.

## Figures and Tables

**Figure 1 ijms-24-10343-f001:**
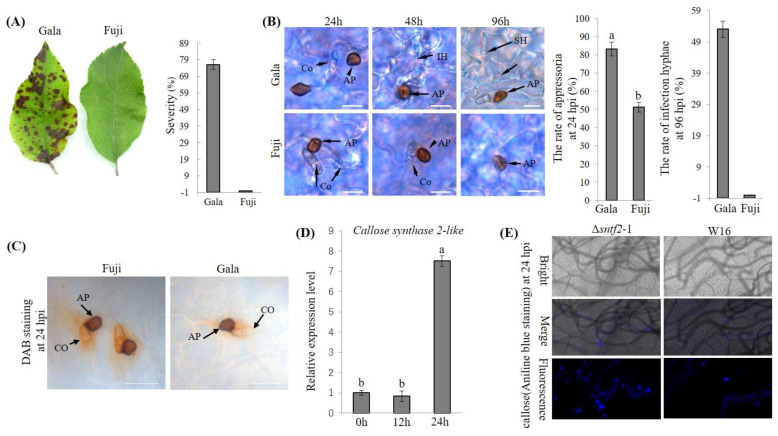
Phenotypic features of *C. gloeosporioides* resistance on Fuji leaves. (**A**) Pathogenicity of *C. gloeosporioides* on Gala and Fuji leaves. Disease severity was determined at 4 dpi. Each experiment was repeated three times, with 10 leaves used per replicate. (**B**) Infection structures of the W16 at 24, 48, and 96 h post–inoculation (hpi). The rate of appressoria and infection hyphae formation of W16 were detected. Lowercase letters indicate significant differences (*p* < 0.01). Co, conidia; Ap, appressorium; IH, infection hyphae; PH, primary hyphae; SH, secondary hyphal. Bar = 10 μm. (**C**) DAB staining was used to detect H_2_O_2_ accumulation. Oxidation of DAB results in the formation of a yellow polymer, which becomes deposited at the site of H_2_O_2_ accumulation. (**D**) qRT–PCR analysis of *Callose synthase 2*–*like* expression in Fuji leaves at 0, 12, and 24 hpi. *MdUBQ* was used as the reference gene. Results are presented as the average fold of values from three independent experiments compared with those from samples at 0 hpi. Lowercase letters indicate significant differences (*p* < 0.01). (**E**) Aniline blue staining to observe callose deposition on apple leaves at 24 hpi.

**Figure 2 ijms-24-10343-f002:**
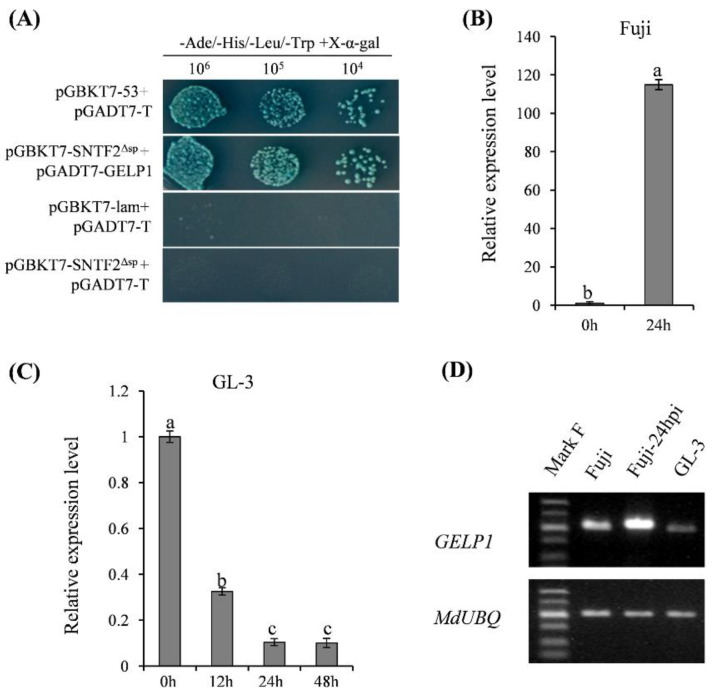
Interaction and expression analysis of *GELP1*. (**A**) Confirmation of the Sntf2–GELP1 interaction using Y2H assays. (**B**) qRT–PCR analysis of *GELP1* expression in Fuji leaves and GL–3 tissues. (**C**) *GELP1* expression during infection. Results are presented as the average fold of values from three independent experiments compared with those of samples at 0 hpi. Lowercase letters indicate significant differences (*p* < 0.01). (**D**) RT–PCR analysis of *GELP1* expression in Fuji and GL–3, *MdUBQ* was used as a control.

**Figure 3 ijms-24-10343-f003:**
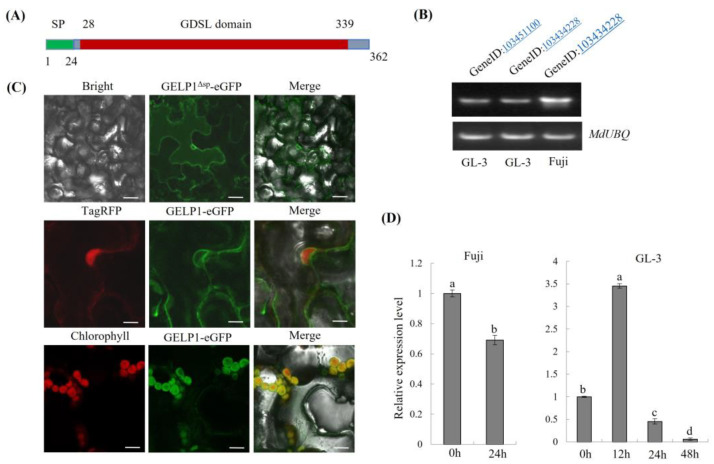
GELP1 localization and GELP2 expression analysis. (**A**) GELP1 was predicted to contain a signal peptide, a GDSL–like domain using Signal Pand InterPro 85.0. (**B**) qRT–PCR and RT–PCR analysis of the expression of *GELP2* in Fuji and GL–3 leaves. Results are presented as the average fold of values from three independent experiments. (**C**) Localization of GELP1 (with or without SP) in *Nicotianabenthamiana*. eGFP protein was used as a control. Bar = 10 μm. (**D**) Relative expression levels of *GELP1* and *GELP2* in Fuji and GL–3 from 0 to 48 hpi. Lowercase letters indicate significant differences (*p* < 0.01).

**Figure 4 ijms-24-10343-f004:**
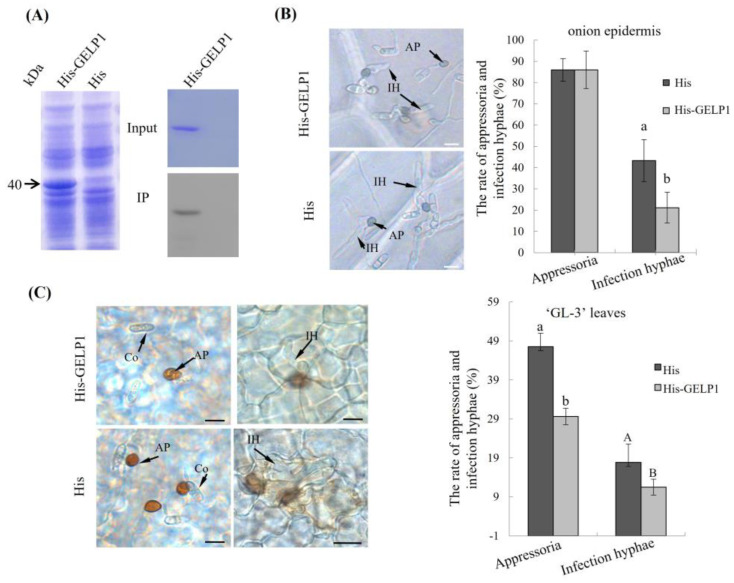
Antimicrobial activity of recombinant GELP1 proteins against *C. gloeosporioides*. (**A**) Coomassie blue–stained gel of purified recombinant His:GELP1 and Western blots analysis. Recombinant His:GELP1 proteins (3 μg) were added to 10 μL of a W16 spore suspension (1 × 10^5^ spores/mL). (**B**) Samples were incubated on onion epidermis for 24 h at 25 °C. (**C**) Samples were incubated on GL–3 leaves for 24 and 48 h at 25 °C. His expression vector was used as a mock control. The rate of appressoria and infection hyphal formation of W16 was detected. Each experiment was repeated three times. Lowercase letters indicate significant differences (*p* < 0.01).

**Figure 5 ijms-24-10343-f005:**
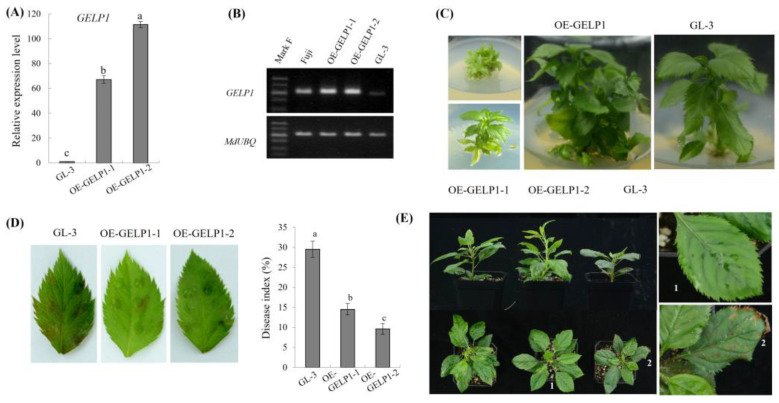
*GELP1* overexpression increases resistance to *C. gloeosporioides* in GL–3. (**A**) Relative expression levels of *GELP1* in transgenic lines using qRT–PCR. The vertical axis represents the relative fold-changes of transcripts compared with leaves of the GL–3 sample. Apple leaf *MdUBQ* was used as the reference gene. Error bars represent standard deviations. Lowercase letters indicate significant differences (*p* < 0.01). (**B**) Analysis of *GELP1* expression in Fuji, GL–3, and transgenic lines leaves by qRT–PCR. (**C**) *GELP1* overexpression lines and GL–3 were cultured on Murashige and Skoog (MS) medium. (**D**) Susceptibility of *GELP1*-overexpression lines 72 h after W16 inoculation. Susceptibility was evaluated based on the disease index. Three replicates were performed for each experiment, with six leaves for each replicate. (**E**) *GELP1* overexpression lines and GL–3 were cultured in a seedling bowl, and pathogenesis was observed after 72 h inoculation.

**Figure 6 ijms-24-10343-f006:**
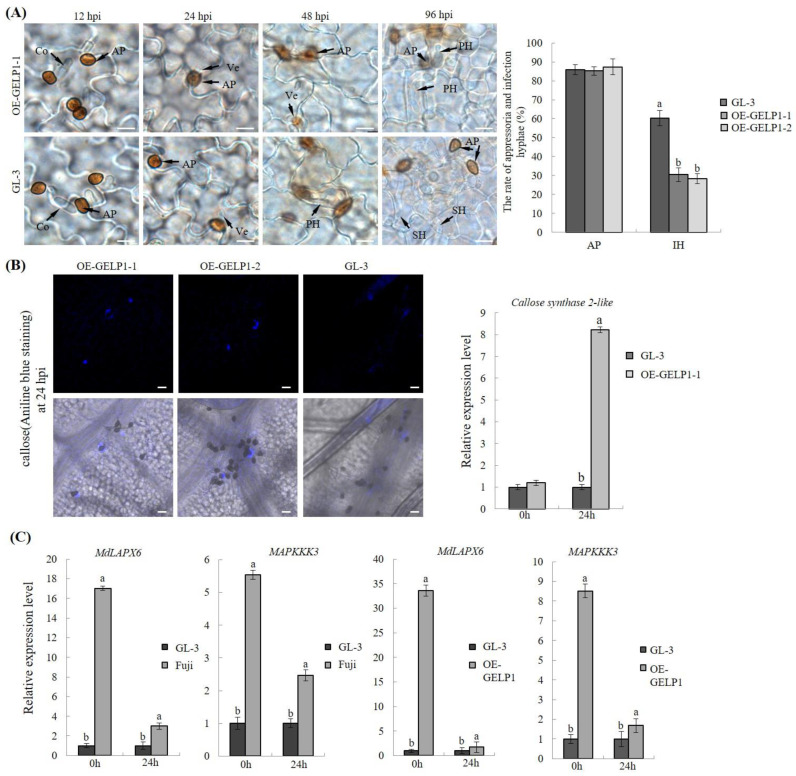
Overexpression of *GELP1* suppressed the invasion of *C. gloeosporioides*. (**A**) Formation of W16 infection hyphae in GL–3 and OE-GELP1 leaves during the infection. Values are presented as means ± SEs. Lowercase letters represent significant differences (*p* < 0.01). Co, conidia; Ap, appressorium; Ve, infection vesicle; PH, primary hyphae; Bar = 10 μm. (**B**) Aniline blue staining to observe callose deposition on leaves at 24 hpi. qRT–PCR analysis the expression of *Callose synthase 2-like* in GL-3 and OE–GELP1 leaves at 0 and 24 hpi. (**C**) Relative expression levels of *MdLAPX6* and *MAPKKK3* in OE–GELP1, GL–3 and Fuji at 0 and 24 hpi. The vertical axis represents the relative fold-changes of transcripts compared with the GL–3 sample. *MdUBQ* was used as the reference gene.

**Figure 7 ijms-24-10343-f007:**
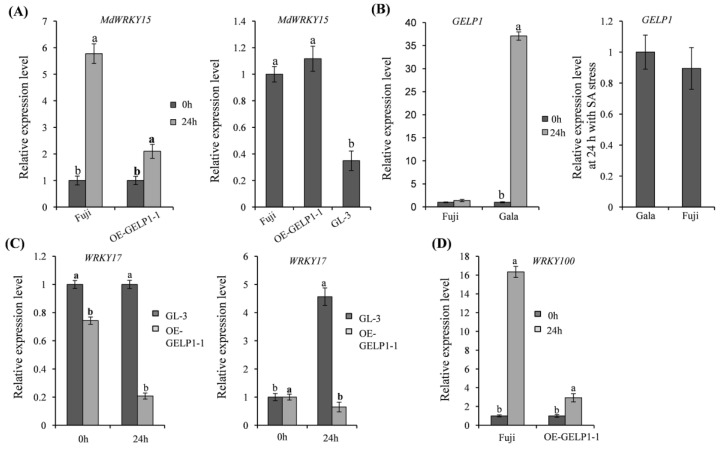
Relative expression levels of *WRKYs* genes. (**A**) *MdWRKY15* expression in OE-GELP1, Fuji, and GL–3 by qRT-PCR analysis. (**B**) qRT–PCR analysis of *GELP1* expression under the salicylic acid (SA, 0.1 mM) stress. (**C**) qRT–PCR analysis of *WRKY17* expression in OE–GELP1 and GL–3. (**D**) qRT–PCR analysis of *WRKY100* expression in OE–GELP1 and Fuji. The vertical axis represents the relative fold–changes of transcripts compared with GL–3 or 0 hpi samples. *MdUBQ* was used as the reference gene. Lowercase letters indicate significant differences (*p* < 0.01).

## Data Availability

The data presented in this study are available in the article.

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
