# Peer review of "GDSL Esterase/Lipase GELP1 Involved in the Defense of Apple Leaves against Colletotrichum gloeosporioides Infection"

_ijms, 2023, doi:10.3390/ijms241210343_

Round 1

Reviewer 1 Report

Zhirui Ji and colleagues present an article on the identification and characterization of enzymes related to the infection of apple leaves with Colletotrichum gloeosporioides by comparing differences between resistant and susceptible varieties. In general the article contains interesting information for people working in this field and is well written. However, the presented results are in my opinion not fully supporting the conclusions. Below are specific comments for consideration.

Chapter 2.1, Figure 1: Please consider including also a merged image in 1E to better follow aniline blue staining.

Chapters 2.2 and 2.3: Authors state that screening of the Fuji Y2H library revealed a potential interacting protein of Sntf2 that encodes a GDSL lipase/esterase like protein named GELP1. What other interacting proteins where identified? What about the complementary experiments using Gala leaves? Additional controls and also a more detailed description are necessary in this experiment. In the same chapter, on Line 133, and in the Chapter 2.3, authors should carefully interpret these results, as both a positive and a negative transcriptional response could also imply a participation in immune defense response. The differences in the expression of GELP1 and GELP2 is very interesting and should be further highlighted in the manuscript.

Figure 3C needs a more detailed description in the text. Controls should be better described and the experiments should also be quantified.

Chapter 2.4: Authors perform an experiment using purified GELP1 and monitor germination and hyphal development. Why was this experiment performed in the first place? Do the authors imply that this protein is secreted and acts extracellularly? This contradicts the localization shown in Figure 3C. More experimental evidence is necessary to ensure that the studied proteins are indeed secreted. Why the authors used onion epidermal cells? Please elaborate.

Chapter 2.5 and 2.6: Please include more details concerning the components of the MAPK pathway that were monitored. Certain observations are based on experiments on different fungi and different hosts. Why do the authors assume that these kinases would have a conserved function in their system? Please elaborate further. More controls should be added to support all that.

Line 199: Authors state that “Compared with GL-3, the transgenic lines showed a similar phenotype on Murashige and Skoog Medium.”  Please explain briefly why these media are and why they were used.

Author Response

Dear Professor:

Thank you very much for your valuable feedback. I have made changes to your comments. The details of the changes are described in the attached document! Many thanks!

Reviewer 2 Report

I congratulate the authors for their work and manuscript. Manuscript ID: ijms-2385748 "GDSL esterase/lipase GELP1 involved in the defense of apple leaves against Colletotrichum gloeosporioides infection" by Ji and collaborators provide the identification and extensive examination of the role of this enzyme during infection leading to Glomerella leaf spot of apple. The authors examined the effect on the fungal germination on plant tissue, its effect on gene expression, and hormonal modulation. The work is very complete and established a solid base for future examinations into its precise molecular mechanisms leading to disease resistance. The manuscript is very well written and the figures are clear and self-explicative. Statistical analysis is well described in the Methods and presented in the figures. I will suggest acceptance for publication however I have just minor comments the authors can work on preparing the final version, especially figure citations as I mentioned below.

Line 182: Review the sentence and check if Clonorchis epididymis should be in italics.

Please review figure numbering in the text. It seems starting from Figure 5 there is a mismatch between the citation and the figures. For example, the text mentions Fig. 5D but the figure does not show panel D.

Line 417: Review for clarity (...180oC per minute...).

English language is fine. Minor edits should be done by the editing team.

Author Response

(The authors gave the same response as above.)

Round 2

Reviewer 1 Report

The authors have successfully responded to some comments of the initial review and the revised manuscript has been improved. Please also consider including in the main text, or as a supplement, the MAPK signal transduction pathway sketch, along with some of the information provided in the response to the reviewer's comments. I have no further suggestions for the authors.

Author Response

Dear Professor,

Many thanks for your valuable feedback. Based on your feedback, I revised the paper again. Details of the changes are found in the attached document! Lots of thanks!
